# STOCHASTIC OPTIMIZATION OF SORTING NETWORKS VIA CONTINUOUS RELAXATIONS

**Aditya Grover**\*, **Eric Wang**\*, **Aaron Zweig & Stefano Ermon**
Computer Science Department
Stanford University
`{adityag,ejwang,azweig,ermon}@cs.stanford.edu`

## ABSTRACT

Sorting input objects is an important step in many machine learning pipelines. However, the sorting operator is non-differentiable with respect to its inputs, which prohibits end-to-end gradient-based optimization. In this work, we propose NeuralSort, a general-purpose continuous relaxation of the output of the sorting operator from permutation matrices to the set of *unimodal* row-stochastic matrices, where every row sums to one and has a distinct $\arg\max$. This relaxation permits straight-through optimization of any computational graph involve a sorting operation. Further, we use this relaxation to enable gradient-based stochastic optimization over the combinatorially large space of permutations by deriving a reparameterized gradient estimator for the Plackett-Luce family of distributions over permutations. We demonstrate the usefulness of our framework on three tasks that require learning semantic orderings of high-dimensional objects, including a fully differentiable, parameterized extension of the $k$-nearest neighbors algorithm.

## 1 INTRODUCTION

Learning to automatically sort objects is useful in many machine learning applications, such as top-$k$ multi-class classification (Berrada et al., 2018), ranking documents for information retrieval (Liu et al., 2009), and multi-object target tracking in computer vision (Bar-Shalom & Li, 1995). Such algorithms typically require learning informative representations of complex, high-dimensional data, such as images, before sorting and subsequent downstream processing. For instance, the $k$-nearest neighbors image classification algorithm, which orders the neighbors based on distances in the canonical pixel basis, can be highly suboptimal for classification (Weinberger et al., 2006). Deep neural networks can instead be used to learn representations, but these representations cannot be optimized end-to-end for a downstream sorting-based objective, since the sorting operator is not differentiable with respect to its input.

In this work, we seek to remedy this shortcoming by proposing NeuralSort, a continuous relaxation to the sorting operator that is differentiable almost everywhere with respect to the inputs. The output of any sorting algorithm can be viewed as a permutation matrix, which is a square matrix with entries in $\{0, 1\}$ such that every row and every column sums to 1. Instead of a permutation matrix, NeuralSort returns a *unimodal* row-stochastic matrix. A unimodal row-stochastic matrix is defined as a square matrix with positive real entries, where each row sums to 1 and has a distinct $\arg\max$. All permutation matrices are unimodal row-stochastic matrices. NeuralSort has a temperature knob that controls the degree of approximation, such that in the limit of zero temperature, we recover a permutation matrix that sorts the inputs. Even for a non-zero temperature, we can efficiently project any unimodal matrix to the desired permutation matrix via a simple row-wise $\arg\max$ operation. Hence, NeuralSort is also suitable for efficient *straight-through gradient optimization* (Bengio et al., 2013), which requires "exact" permutation matrices to evaluate learning objectives.

As the second primary contribution, we consider the use of NeuralSort for stochastic optimization over permutations. In many cases, such as latent variable models, the permutations may be latent but directly influence observed behavior, *e.g.*, utility and choice models are often expressed as distributions over permutations which govern the observed decisions of agents (Regenwetter et al.,

---

\*Equal contribution

2006; Chierichetti et al., 2018). By learning distributions over unobserved permutations, we can account for the uncertainty in these permutations in a principled manner. However, the challenge with stochastic optimization over discrete distributions lies in gradient estimation with respect to the distribution parameters. Vanilla REINFORCE estimators are impractical for most cases, or necessitate custom control variates for low-variance gradient estimation (Glasserman, 2013).

In this regard, we consider the Plackett-Luce (PL) family of distributions over permutations (Plackett, 1975; Luce, 1959). A common modeling choice for ranking models, the PL distribution is parameterized by $n$ scores, with its support defined over the symmetric group consisting of $n!$ permutations. We derive a reparameterizable sampler for stochastic optimization with respect to this distribution, based on Gumbel perturbations to the $n$ (log-)scores. However, the reparameterized sampler requires sorting these perturbed scores, and hence the gradients of a downstream learning objective with respect to the scores are not defined. By using NeuralSort instead, we can approximate the objective and obtain well-defined reparameterized gradient estimates for stochastic optimization.

Finally, we apply NeuralSort to tasks that require us to learn semantic orderings of complex, high-dimensional input data. First, we consider sorting images of handwritten digits, where the goal is to learn to sort images by their unobserved labels. Our second task extends the first one to quantile regression, where we want to estimate the median (50-th percentile) of a set of handwritten numbers. In addition to identifying the index of the median image in the sequence, we need to learn to map the inferred median digit to its scalar representation. In the third task, we propose an algorithm that learns a basis representation for the $k$-nearest neighbors (kNN) classifier in an end-to-end procedure. Because the choice of the $k$ nearest neighbors requires a non-differentiable sorting, we use NeuralSort to obtain an approximate, differentiable surrogate. On all tasks, we observe significant empirical improvements due to NeuralSort over the relevant baselines and competing relaxations to permutation matrices.

## 2 PRELIMINARIES

An $n$-dimensional permutation $\mathbf{z} = [z_1, z_2, \ldots, z_n]^T$ is a list of unique indices $\{1, 2, \ldots, n\}$. Every permutation $\mathbf{z}$ is associated with a permutation matrix $P_\mathbf{z} \in \{0, 1\}^{n \times n}$ with entries given as:

$$P_\mathbf{z}[i, j] = \begin{cases} 1 \text{ if } j = z_i \\ 0 \text{ otherwise.} \end{cases}$$

Let $\mathcal{Z}_n$ denote the set of all $n!$ possible permutations in the symmetric group. We define the $\texttt{sort} : \mathbb{R}^n \rightarrow \mathcal{Z}_n$ operator as a mapping of $n$ real-valued inputs to a permutation corresponding to a descending ordering of these inputs. *E.g.*, if the input vector $\mathbf{s} = [9, 1, 5, 2]^T$, then $\texttt{sort}(\mathbf{s}) = [1, 3, 4, 2]^T$ since the largest element is at the first index, second largest element is at the third index and so on. In case of ties, elements are assigned indices in the order they appear. We can obtain the sorted vector simply via $P_{\texttt{sort}(\mathbf{s})}\mathbf{s}$.

### 2.1 PLACKETT-LUCE DISTRIBUTIONS

The family of Plackett-Luce distributions over permutations is best described via a generative process: Consider a sequence of $n$ items, each associated with a canonical index $i = 1, 2, \ldots, n$. A common assumption in ranking models is that the underlying generating process for any observed permutation of $n$ items satisfies Luce's choice axiom (Luce, 1959). Mathematically, this axiom defines the 'choice' probability of an item with index $i$ as: $q(i) \propto s_i$ where $s_i > 0$ is interpreted as the score of item with index $i$. The normalization constant is given by $Z = \sum_{i \in \{1, 2, \ldots, n\}} s_i$.

If we choose the $n$ items one at a time (without replacement) based on these choice probabilities, we obtain a discrete distribution over all possible permutations. This distribution is referred to as the Plackett-Luce (PL) distribution, and its probability mass function for any $\mathbf{z} \in \mathcal{Z}_n$ is given by:

$$q(\mathbf{z}|\mathbf{s}) = \frac{s_{z_1}}{Z} \frac{s_{z_2}}{Z - s_{z_1}} \cdots \frac{s_{z_n}}{Z - \sum_{i=1}^{n-1} s_{z_i}} \tag{1}$$

where $\mathbf{s} = \{s_1, s_2, \ldots, s_n\}$ is the vector of scores parameterizing this distribution (Plackett, 1975).

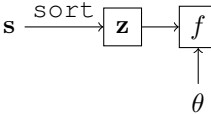

Figure 1: Stochastic computation graphs with a deterministic node $\mathbf{z}$ corresponding to the output of a sort operator applied to the scores $\mathbf{s}$.

## 2.2 STOCHASTIC COMPUTATION GRAPHS

The abstraction of *stochastic computation graphs* (SCG) compactly specifies the forward value and the backward gradient computation for computational circuits. An SCG is a directed acyclic graph that consists of three kinds of nodes: *input* nodes which specify external inputs (including parameters), *deterministic* nodes which are deterministic functions of their parents, and *stochastic* nodes which are distributed conditionally on their parents. See Schulman et al. (2015) for a review.

To define gradients of an objective function with respect to any node in the graph, the chain rule necessitates that the gradients with respect to the intermediate nodes are well-defined. This is not the case for the sort operator. In Section 3, we propose to extend stochastic computation graphs with nodes corresponding to a relaxation of the deterministic sort operator. In Section 4, we further use this relaxation to extend computation graphs to include stochastic nodes corresponding to distributions over permutations. The proofs of all theoretical results in this work are deferred to Appendix B.

## 3 NEURALSORT: THE RELAXED SORTING OPERATOR

Our goal is to optimize training objectives involving a sort operator with gradient-based methods. Consider the optimization of objectives written in the following form:

$$L(\theta, \mathbf{s}) = f(P_{\mathbf{z}}; \theta) \tag{2}$$
$$\text{where } \mathbf{z} = \text{sort}(\mathbf{s}).$$

Here, $\mathbf{s} \in \mathbb{R}^n$ denotes a vector of $n$ real-valued scores, $\mathbf{z}$ is the permutation that (deterministically) sorts the scores $\mathbf{s}$, and $f(\cdot)$ is an arbitrary function of interest assumed to be differentiable w.r.t a set of parameters $\theta$ and $\mathbf{z}$. For example, in a ranking application, these scores could correspond to the inferred relevances of $n$ webpages and $f(\cdot)$ could be a ranking loss. Figure 1 shows the stochastic computation graph corresponding to the objective in Eq. 2. We note that this could represent part of a more complex computation graph, which we skip for ease of presentation while maintaining the generality of the scope of this work.

While the gradient of the above objective w.r.t. $\theta$ is well-defined and can be computed via standard backpropogation, the gradient w.r.t. the scores $\mathbf{s}$ is not defined since the sort operator is not differentiable w.r.t. $\mathbf{s}$. Our solution is to derive a relaxation to the sort operator that leads to a surrogate objective with well-defined gradients. In particular, we seek to use such a relaxation to replace the permutation matrix $P_{\mathbf{z}}$ in Eq. 2 with an approximation $\widehat{P}_{\mathbf{z}}$ such that the surrogate objective $f(\widehat{P}_{\mathbf{z}}; \theta)$ is differentiable w.r.t. the scores $\mathbf{s}$.

The general recipe to relax non-differentiable operators with *discrete* codomains $\mathcal{N}$ is to consider differentiable alternatives that map the input to a larger *continuous* codomain $\mathcal{M}$ with desirable properties. For gradient-based optimization, we are interested in two key properties:

1. The relaxation is continuous everywhere and differentiable (almost-)everywhere with respect to elements in the input domain.
2. There exists a computationally efficient projection from $\mathcal{M}$ back to $\mathcal{N}$.

Relaxations satisfying the first requirement are amenable to automatic differentiation for optimizing stochastic computational graphs. The second requirement is useful for evaluating metrics and losses that necessarily require a discrete output akin to the one obtained from the original, non-relaxed operator. *E.g.*, in straight-through gradient estimation (Bengio et al., 2013; Jang et al., 2017), the

$$\begin{pmatrix} 0 & 1/2 & 1/2 \\ 7/16 & 3/16 & 3/8 \\ 9/16 & 5/16 & 1/8 \end{pmatrix} \qquad \begin{pmatrix} 3/8 & 1/8 & 1/2 \\ 3/4 & 1/4 & 0 \\ 1/4 & 1/2 & 1/4 \end{pmatrix}$$

Figure 2: **Center:** Venn Diagram relationships between permutation matrices ($\mathcal{P}$), doubly-stochastic matrices ($\mathcal{D}$), unimodal row stochastic matrices ($\mathcal{U}$), and row stochastic matrices ($\mathcal{R}$). **Left:** A doubly-stochastic matrix that is not unimodal. **Right**: A unimodal matrix that is not doubly-stochastic.

non-relaxed operator is used for evaluating the learning objective in the forward pass and the relaxed operator is used in the backward pass for gradient estimation.

The canonical example is the $0/1$ loss used for binary classification. While the $0/1$ loss is discontinuous w.r.t. its inputs (real-valued predictions from a model), surrogates such as the logistic and hinge losses are continuous everywhere and differentiable almost-everywhere (property 1), and can give hard binary predictions via thresholding (property 2).

*Note:* For brevity, we assume that the $\arg\max$ operator is applied over a set of elements with a unique maximizer and hence, the operator has well-defined semantics. With some additional bookkeeping for resolving ties, the results in this section hold even if the elements to be sorted are not unique. See Appendix C.

**Unimodal Row Stochastic Matrices.** The `sort` operator maps the input vector to a permutation, or equivalently a permutation matrix. Our relaxation to `sort` is motivated by the geometric structure of permutation matrices. The set of permutation matrices is a subset of *doubly-stochastic matrices*, i.e., a non-negative matrix such that the every row and column sums to one. If we remove the requirement that every column should sum to one, we obtain a larger set of *row stochastic matrices*. In this work, we propose a relaxation to `sort` that maps inputs to an alternate subset of row stochastic matrices, which we refer to as the *unimodal row stochastic matrices*.

**Definition 1** (Unimodal Row Stochastic Matrices). *An $n \times n$ matrix is Unimodal Row Stochastic if it satisfies the following conditions:*

1. ***Non-negativity:*** $U[i, j] \geq 0 \quad \forall i, j \in \{1, 2, \ldots, n\}.$

2. ***Row Affinity:*** $\sum_{j=1}^{n} U[i, j] = 1 \quad \forall i \in \{1, 2, \ldots, n\}.$

3. ***Argmax Permutation:*** *Let* $\mathbf{u}$ *denote an $n$-dimensional vector with entries such that* $u_i = \arg\max_j U[i, j] \quad \forall i \in \{1, 2, \ldots, n\}.$ *Then,* $\mathbf{u} \in \mathcal{Z}_n$, *i.e., it is a valid permuation.*

*We denote $\mathcal{U}_n$ as the set of $n \times n$ unimodal row stochastic matrices.*

All row stochastic matrices satisfy the first two conditions. The third condition is useful for gradient based optimization involving sorting-based losses. The condition provides a straightforward mechanism for extracting a permutation from a unimodal row stochastic matrix via a row-wise $\arg\max$ operation. Figure 2 shows the relationships between the different subsets of square matrices.

**NeuralSort.** Our relaxation to the `sort` operator is based on a standard identity for evaluating the sum of the $k$ largest elements in any input vector.

**Lemma 2.** *[Lemma 1 in Ogryczak & Tamir (2003)] For an input vector $\mathbf{s} = [s_1, s_2, \ldots, s_n]^T$ that is sorted as $s_{[1]} \geq s_{[2]} \geq \ldots \geq s_{[n]}$, we have the sum of the $k$-largest elements given as:*

$$\sum_{i=1}^{k} s_{[i]} = \min_{\lambda \in \{s_1, s_2, \ldots, s_n\}} \lambda k + \sum_{i=1}^{n} \max(s_i - \lambda, 0). \tag{3}$$

The identity in Lemma 2 outputs the sum of the top-$k$ elements. The $k$-th largest element itself can be recovered by taking the difference of the sum of top-$k$ elements and the top-$(k-1)$ elements.

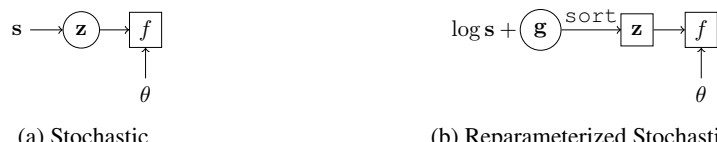

(a) Stochastic            (b) Reparameterized Stochastic

Figure 3: Stochastic computation graphs with stochastic nodes corresponding to permutations. Squares denote deterministic nodes and circles denote stochastic nodes.

**Corollary 3.** *Let $\mathbf{s} = [s_1, s_2, \ldots, s_n]^T$ be a real-valued vector of length $n$. Let $A_{\mathbf{s}}$ denote the matrix of absolute pairwise differences of the elements of $\mathbf{s}$ such that $A_{\mathbf{s}}[i, j] = |s_i - s_j|$. The permutation matrix $P_{sort(\mathbf{s})}$ corresponding to $sort(\mathbf{s})$ is given by:*

$$P_{sort(\mathbf{s})}[i,j] = \begin{cases} 1 \text{ if } j = \arg\max[(n+1-2i)\mathbf{s} - A_{\mathbf{s}}\mathbb{1}] \\ 0 \text{ otherwise} \end{cases} \tag{4}$$

*where $\mathbb{1}$ denotes the column vector of all ones.*

*E.g.*, if we set $i = \lfloor(n+1)/2\rfloor$ then the non-zero entry in the $i$-th row $P_{sort(\mathbf{s})}[i, :]$ corresponds to the element with the minimum sum of (absolute) distance to the other elements. As desired, this corresponds to the median element. The relaxation requires $O(n^2)$ operations to compute $A_{\mathbf{s}}$, as opposed to the $O(n \log n)$ overall complexity for the best known sorting algorithms. In practice however, it is highly parallelizable and can be implemented efficiently on GPU hardware.

The $\arg\max$ operator is non-differentiable which prohibits the direct use of Corollary 3 for gradient computation. Instead, we propose to replace the $\arg\max$ operator with $\operatorname{soft}\max$ to obtain a continuous relaxation $\widehat{P}_{sort(\mathbf{s})}(\tau)$. In particular, the $i$-th row of $\widehat{P}_{sort(\mathbf{s})}(\tau)$ is given by:

$$\widehat{P}_{sort(\mathbf{s})}[i,:](\tau) = \operatorname{soft}\max\left[((n+1-2i)\mathbf{s} - A_{\mathbf{s}}\mathbb{1})/\tau\right] \tag{5}$$

where $\tau > 0$ is a temperature parameter. Our relaxation is continuous everywhere and differentiable almost everywhere with respect to the elements of $\mathbf{s}$. Furthermore, we have the following result.

**Theorem 4.** *Let $\widehat{P}_{sort(\mathbf{s})}$ denote the continuous relaxation to the permutation matrix $P_{sort(\mathbf{s})}$ for an arbitrary input vector $\mathbf{s}$ and temperature $\tau$ defined in Eq. 5. Then, we have:*

1. *Unimodality: $\forall \tau > 0$, $\widehat{P}_{sort(\mathbf{s})}$ is a unimodal row stochastic matrix. Further, let $\mathbf{u}$ denote the permutation obtained by applying $\arg\max$ row-wise to $\widehat{P}_{sort(\mathbf{s})}$. Then, $\mathbf{u} = sort(\mathbf{s})$.*

2. *Limiting behavior: If we assume that the entries of $\mathbf{s}$ are drawn independently from a distribution that is absolutely continuous w.r.t. the Lebesgue measure in $\mathbb{R}$, then the following convergence holds almost surely:*

$$\lim_{\tau \to 0^+} \widehat{P}_{sort(\mathbf{s})}[i,:](\tau) = P_{sort(\mathbf{s})}[i,:] \quad \forall i \in \{1, 2, \ldots, n\}. \tag{6}$$

Unimodality allows for efficient projection of the relaxed permutation matrix $\widehat{P}_{sort(\mathbf{s})}$ to the hard matrix $P_{sort(\mathbf{s})}$ via a row-wise $\arg\max$, *e.g.*, for straight-through gradients. For analyzing limiting behavior, independent draws ensure that the elements of $\mathbf{s}$ are distinct almost surely. The temperature $\tau$ controls the degree of smoothness of our approximation. At one extreme, the approximation becomes tighter as the temperature is reduced. In practice however, the trade-off is in the variance of these estimates, which is typically lower for larger temperatures.

## 4 STOCHASTIC OPTIMIZATION OVER PERMUTATIONS

In many scenarios, we would like the ability to express our uncertainty in inferring a permutation *e.g.*, latent variable models with latent nodes corresponding to permutations. Random variables that assume values corresponding to permutations can be represented via stochastic nodes in the

stochastic computation graph. For optimizing the parameters of such a graph, consider the following class of objectives:

$$L(\theta, \mathbf{s}) = \mathbb{E}_{q(\mathbf{z}|\mathbf{s})} \left[ f(P_{\mathbf{z}}; \theta) \right] \tag{7}$$

where $\theta$ and $\mathbf{s}$ denote sets of parameters, $P_{\mathbf{z}}$ is the permutation matrix corresponding to the permutation $\mathbf{z}$, $q(\cdot)$ is a parameterized distribution over the elements of the symmetric group $\mathcal{Z}_n$, and $f(\cdot)$ is an arbitrary function of interest assumed to be differentiable in $\theta$ and $\mathbf{z}$. The SCG is shown in Figure 3a. In contrast to the SCG considered in the previous section (Figure 1), here we are dealing with a distribution over permutations as opposed to a single (deterministically computed) one.

While such objectives are typically intractable to evaluate exactly since they require summing over a combinatorially large set, we can obtain unbiased estimates efficiently via Monte Carlo. Monte Carlo estimates of gradients w.r.t. $\theta$ can be derived simply via linearity of expectation. However, the gradient estimates w.r.t. $\mathbf{s}$ cannot be obtained directly since the sampling distribution depends on $\mathbf{s}$. The *REINFORCE gradient estimator* (Glynn, 1990; Williams, 1992; Fu, 2006) uses the fact that $\nabla_{\mathbf{s}} q(\mathbf{z}|\mathbf{s}) = q(\mathbf{z}|\mathbf{s}) \nabla_{\mathbf{s}} \log q(\mathbf{z}|\mathbf{s})$ to derive the following Monte Carlo gradient estimates:

$$\nabla_{\mathbf{s}} L(\theta, \mathbf{s}) = \mathbb{E}_{q(\mathbf{z}|\mathbf{s})} \left[ f(P_{\mathbf{z}}; \theta) \nabla_{\mathbf{s}} \log q(\mathbf{z}|\mathbf{s}) \right] + \mathbb{E}_{q(\mathbf{z}|\mathbf{s})} \left[ \nabla_{\mathbf{s}} f(P_{\mathbf{z}}; \theta) \right]. \tag{8}$$

## 4.1 Reparameterized gradient estimators for PL distributions

REINFORCE gradient estimators typically suffer from high variance (Schulman et al., 2015; Glasserman, 2013). *Reparameterized samplers* provide an alternate gradient estimator by expressing samples from a distribution as a *deterministic* function of its parameters and a fixed source of randomness (Kingma & Welling, 2014; Rezende et al., 2014; Titsias & Lázaro-Gredilla, 2014). Since the randomness is from a fixed distribution, Monte Carlo gradient estimates can be derived by pushing the gradient operator inside the expectation (via linearity). In this section, we will derive a reparameterized sampler and gradient estimator for the Plackett-Luce (PL) family of distributions.

Let the score $s_i$ for an item $i \in \{1, 2, \ldots, n\}$ be an unobserved random variable drawn from some underlying score distribution (Thurstone, 1927). Now for each item, we draw a score from its corresponding score distribution. Next, we generate a permutation by applying the deterministic `sort` operator to these $n$ *randomly sampled* scores. Interestingly, prior work has shown that the resulting distribution over permutations corresponds to a PL distribution if and only if the scores are sampled independently from Gumbel distributions with identical scales.

**Proposition 5.** *[adapted from Yellott Jr (1977)] Let $\mathbf{s}$ be a vector of scores for the $n$ items. For each item $i$, sample $g_i \sim \text{Gumbel}(0, \beta)$ independently with zero mean and a fixed scale $\beta$. Let $\tilde{\mathbf{s}}$ denote the vector of Gumbel perturbed log-scores with entries such that $\tilde{s}_i = \beta \log s_i + g_i$. Then:*

$$q(\tilde{s}_{z_1} \geq \cdots \geq \tilde{s}_{z_n}) = \frac{s_{z_1}}{Z} \frac{s_{z_2}}{Z - s_{z_1}} \cdots \frac{s_{z_n}}{Z - \sum_{i=1}^{n-1} s_{z_i}}. \tag{9}$$

For ease of presentation, we assume $\beta = 1$ in the rest of this work. Proposition 5 provides a method for sampling from PL distributions with parameters $\mathbf{s}$ by adding Gumbel perturbations to the log-scores and applying the `sort` operator to the perturbed log-scores. This procedure can be seen as a reparameterization trick that expresses a sample from the PL distribution as a deterministic function of the scores and a fixed source of randomness (Figure 3b). Letting $\mathbf{g}$ denote the vector of i.i.d. Gumbel perturbations, we can express the objective in Eq. 7 as:

$$L(\theta, \mathbf{s}) = \mathbb{E}_{\mathbf{g}} \left[ f(P_{\text{sort}(\log \mathbf{s} + \mathbf{g})}; \theta) \right]. \tag{10}$$

While the reparameterized sampler removes the dependence of the expectation on the parameters $\mathbf{s}$, it introduces a `sort` operator in the computation graph such that the overall objective is non-differentiable in $\mathbf{s}$. In order to obtain a differentiable surrogate, we approximate the objective based on the NeuralSort relaxation to the `sort` operator:

$$\mathbb{E}_{\mathbf{g}} \left[ f(P_{\text{sort}(\log \mathbf{s} + \mathbf{g})}; \theta) \right] \approx \mathbb{E}_{\mathbf{g}} \left[ f(\widehat{P}_{\text{sort}(\log \mathbf{s} + \mathbf{g})}; \theta) \right] := \widehat{L}(\theta, \mathbf{s}). \tag{11}$$

Accordingly, we get the following reparameterized gradient estimates for the approximation:

$$\nabla_{\mathbf{s}} \widehat{L}(\theta, \mathbf{s}) = \mathbb{E}_{\mathbf{g}} \left[ \nabla_{\mathbf{s}} f(\widehat{P}_{\text{sort}(\log \mathbf{s} + \mathbf{g})}; \theta) \right] \tag{12}$$

which can be estimated efficiently via Monte Carlo because the expectation is with respect to a distribution that does not depend on $\mathbf{s}$.

## 5 DISCUSSION AND RELATED WORK

The problem of learning to rank documents based on relevance has been studied extensively in the context of information retrieval. In particular, *listwise* approaches learn functions that map objects to scores. Much of this work concerns the PL distribution: the RankNet algorithm (Burges et al., 2005) can be interpreted as maximizing the PL likelihood of pairwise comparisons between items, while the ListMLE ranking algorithm in Xia et al. (2008) extends this with a loss that maximizes the PL likelihood of ground-truth permutations directly. The differentiable *pairwise* approaches to ranking, such as Rigutini et al. (2011), learn to approximate the comparator between pairs of objects. Our work considers a generalized setting where sorting based operators can be inserted anywhere in computation graphs to extend traditional pipelines *e.g.*, kNN.

Prior works have proposed relaxations of permutation matrices to the Birkhoff polytope, which is defined as the convex hull of the set of permutation matrices a.k.a. the set of doubly-stochastic matrices. A doubly-stochastic matrix is a permutation matrix iff it is orthogonal and continuous relaxations based on these matrices have been used previously for solving NP-complete problems such as seriation and graph matching (Fogel et al., 2013; Fiori et al., 2013; Lim & Wright, 2014). Adams & Zemel (2011) proposed the use of the Sinkhorn operator to map any *square matrix* to the Birkhoff polytope. They interpret the resulting doubly-stochastic matrix as the marginals of a distribution over permutations. Mena et al. (2018) propose an alternate method where the square matrix defines a latent distribution over the doubly-stochastic matrices themselves. These distributions can be sampled from by adding elementwise Gumbel perturbations. Linderman et al. (2018) propose a *rounding* procedure that uses the Sinkhorn operator to directly sample matrices near the Birkhoff polytope. Unlike Mena et al. (2018), the resulting distribution over matrices has a tractable density. In practice, however, the approach of Mena et al. (2018) performs better and will be the main baseline we will be comparing against in our experiments in Section 6.

As discussed in Section 3, NeuralSort maps permutation matrices to the set of unimodal row-stochastic matrices. For the stochastic setting, the PL distribution permits efficient sampling, exact and tractable density estimation, making it an attractive choice for several applications, *e.g.*, variational inference over latent permutations. Our reparameterizable sampler, while also making use of the Gumbel distribution, is based on a result unique to the PL distribution (Proposition 5).

The use of the Gumbel distribution for defining continuous relaxations to discrete distributions was first proposed concurrently by Jang et al. (2017) and Maddison et al. (2017) for categorical variables, referred to as Gumbel-Softmax. The number of possible permutations grow factorially with the dimension, and thus any distribution over $n$-dimensional permutations can be equivalently seen as a distribution over $n!$ categories. Gumbel-softmax does not scale to a combinatorially large number of categories (Kim et al., 2016; Mussmann et al., 2017), necessitating the use of alternate relaxations, such as the one considered in this work.

## 6 EXPERIMENTS

We refer to the two approaches proposed in Sections 3, 4 as Deterministic NeuralSort and Stochastic NeuralSort, respectively. For additional hyperparameter details and analysis, see Appendix D.

### 6.1 SORTING HANDWRITTEN NUMBERS

**Dataset.** We first create the *large-MNIST* dataset, which extends the MNIST dataset of handwritten digits. The dataset consists of multi-digit images, each a concatenation of 4 randomly selected individual images from MNIST, *e.g.*,  is one such image in this dataset. Each image is associated with a real-valued label, which corresponds to its concatenated MNIST labels, *e.g.*, the label of  is 1810. Using the large-MNIST dataset, we finally create a dataset of sequences. Every sequence is this dataset consists of $n$ randomly sampled large-MNIST images.

**Setup.** Given a dataset of sequences of large-MNIST images, our goal is to learn to predict the permutation that sorts the labels of the sequence of images, given a training set of ground-truth permutations. Figure 4 (Task 1) illustrates this task on an example sequence of $n = 5$ large-MNIST images. This task is a challenging extension of the one considered by Mena et al. (2018) in sorting scalars, since it involves learning the semantics of high-dimensional objects prior to sorting. A

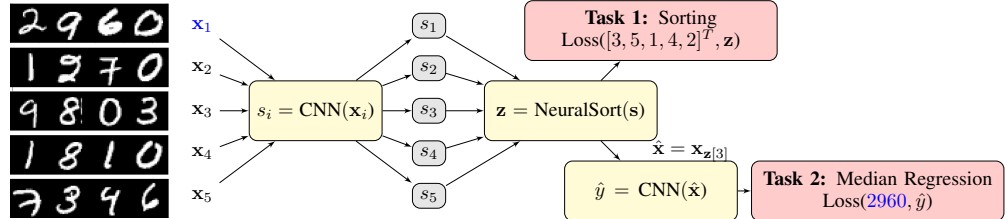

Figure 4: Sorting and quantile regression. The model is trained to sort sequences of $n = 5$ large-MNIST images $\mathbf{x}_1, \mathbf{x}_2, \ldots, \mathbf{x}_5$ (**Task 1**) and regress the median value (**Task 2**). In the above example, the ground-truth permutation that sorts the input sequence from largest to smallest is $[3, 5, 1, 4, 2]^T$, 9803 being the largest and 1270 the smallest. Blue illustrates the true median image $\mathbf{x}_1$ with ground-truth sorted index 3 and value 2960.

Table 1: Average sorting accuracy on the test set. First value is proportion of permutations correctly identified; value in parentheses is the proportion of individual element ranks correctly identified.

| Algorithm | $n = 3$ | $n = 5$ | $n = 7$ | $n = 9$ | $n = 15$ |
|---|---|---|---|---|---|
| Vanilla RS | 0.467 (0.801) | 0.093 (0.603) | 0.009 (0.492) | 0. (0.113) | 0. (0.067) |
| Sinkhorn | 0.462 (0.561) | 0.038 (0.293) | 0.001 (0.197) | 0. (0.143) | 0. (0.078) |
| Gumbel-Sinkhorn | 0.484 (0.575) | 0.033 (0.295) | 0.001 (0.189) | 0. (0.146) | 0. (0.078) |
| Deterministic NeuralSort | **0.930 (0.951)** | **0.837 (0.927)** | 0.738 (**0.909**) | **0.649 (0.896)** | 0.386 (0.857) |
| Stochastic NeuralSort | 0.927 (0.950) | 0.835 (0.926) | **0.741 (0.909)** | 0.646 (0.895) | **0.418 (0.862)** |

good model needs to learn to dissect the individual digits in an image, rank these digits, and finally, compose such rankings based on the digit positions within an image. The available supervision, in the form of the ground-truth permutation, is very weak compared to a classification setting that gives direct access to the image labels.

**Baselines.** All baselines use a CNN that is shared across all images in a sequence to map each large-MNIST image to a feature space. The *vanilla row-stochastic (RS)* baseline concatenates the CNN representations for $n$ images into a single vector that is fed into a multilayer perceptron that outputs $n$ multiclass predictions of the image probabilities for each rank. The *Sinkhorn* and *Gumbel-Sinkhorn* baselines, as discussed in Section 5, use the Sinkhorn operator to map the stacked CNN representations for the $n$ objects into a doubly-stochastic matrix. For all methods, we minimized the cross-entropy loss between the predicted matrix and the ground-truth permutation matrix.

**Results.** Following Mena et al. (2018), our evaluation metric is the the proportion of correctly predicted permutations on a test set of sequences. Additionally, we evaluate the proportion of individual elements ranked correctly. Table 1 demonstrates that the approaches based on the proposed sorting relaxation significantly outperform the baseline approaches for all $n$ considered. The performance of the deterministic and stochastic variants are comparable. The vanilla RS baseline performs well in ranking individual elements, but is not good at recovering the overall square matrix.

We believe the poor performance of the Sinkhorn baselines is partly because these methods were designed and evaluated for *matchings*. Like the output of `sort`, matchings can also be represented as permutation matrices. However, distributions over matchings need not satisfy Luce's choice axiom or imply a total ordering, which could explain the poor performance on the tasks considered.

## 6.2 QUANTILE REGRESSION

**Setup.** In this experiment, we extend the sorting task to regression. Again, each sequence contains $n$ large-MNIST images, and the regression target for each sequence is the 50-th quantile (*i.e.*, the median) of the $n$ labels of the images in the sequence. Figure 4 (Task 2) illustrates this task on an example sequence of $n = 5$ large-MNIST images, where the goal is to output the third largest label. The design of this task highlights two key challenges since it explicitly requires learning both a suitable representation for sorting high-dimensional inputs and a secondary function that approximates the label itself (regression). Again, the supervision available in the form of the label of only a single image at an arbitrary and unknown location in the sequence is weak.

Table 2: Test mean squared error ($\times 10^{-4}$) and $R^2$ values (in parenthesis) for quantile regression.

| Algorithm | $n = 5$ | $n = 9$ | $n = 15$ |
|---|---|---|---|
| Constant (Simulated) | 356.79 (0.00) | 227.31 (0.00) | 146.94 ( 0.00) |
| Vanilla NN | 1004.70 (0.85) | 699.15 (0.82) | 562.97 (0.79) |
| Sinkhorn | 343.60 (0.25) | 231.87 (0.19) | 156.27 (0.04) |
| Gumbel-Sinkhorn | 344.28 (0.25) | 232.56 (0.23) | 157.34 (0.06) |
| Deterministic NeuralSort | 45.50 (**0.95**) | 34.98 (**0.94**) | 34.78 (**0.92**) |
| Stochastic NeuralSort | **33.80** (0.94) | **31.43** (0.93) | **29.34** (0.90) |

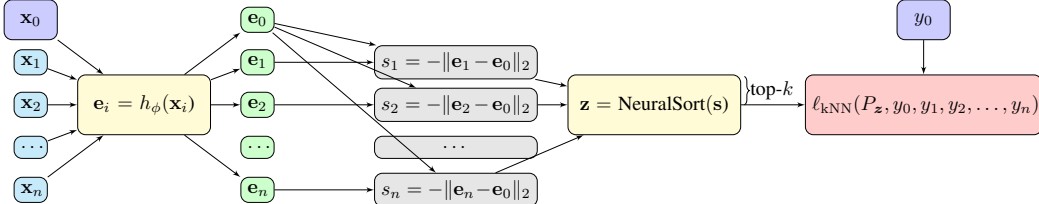

Figure 5: Differentiable kNN. The model is trained such that the representations $\mathbf{e}_i$ for the training points $\{\mathbf{x}_1, \ldots, \mathbf{x}_n\}$ that have the same label $y_0$ as $\mathbf{x}_0$ are closer to $\mathbf{e}_0$ (included in top-$k$) than others.

**Baselines.** In addition to *Sinkhorn* and *Gumbel-Sinkhorn*, we design two more baselines. The *Constant* baseline always returns the *median* of the full range of possible outputs, ignoring the input sequence. This corresponds to 4999.5 since we are sampling large-MNIST images uniformly in the range of four-digit numbers. The *vanilla neural net (NN)* baseline directly maps the input sequence of images to a real-valued prediction for the median.

**Results.** Our evaluation metric is the mean squared error (MSE) and $R^2$ on a test set of sequences. Results for $n = \{5, 9, 15\}$ images are shown in Table 2. The Vanilla NN baseline while incurring a large MSE, is competitive on the $R^2$ metric. The other baselines give comparable performance on the MSE metric. The proposed NeuralSort approaches outperform the competing methods on both the metrics considered. The stochastic NeuralSort approach is the consistent best performer on MSE, while the deterministic NeuralSort is slightly better on the $R^2$ metric.

## 6.3 END-TO-END, DIFFERENTIABLE $k$-NEAREST NEIGHBORS

**Setup.** In this experiment, we design a fully differentiable, end-to-end $k$-nearest neighbors (kNN) classifier. Unlike a standard kNN classifier which computes distances between points in a predefined space, we learn a representation of the data points before evaluating the $k$-nearest neighbors.

We are given access to a dataset $\mathcal{D}$ of $(\mathbf{x}, y)$ pairs of standard input data and their class labels respectively. The differentiable kNN algorithm consists of two hyperparameters: the number of training neighbors $n$, the number of top candidates $k$, and the sorting temperature $\tau$. Every sequence of items here consists of a query point $\mathbf{x}$ and a randomly sampled subset of $n$ candidate nearest neighbors from the training set, say $\{\mathbf{x}_1, \mathbf{x}_2, \ldots, \mathbf{x}_n\}$. In principle, we could use the entire training set (excluding the query point) as candidate points, but this can hurt the learning both computationally and statistically. The query points are randomly sampled from the train/validation/test sets as appropriate but the nearest neighbors are always sampled from the training set. The loss function optimizes for a representation space $h_\phi(\cdot)$ (*e.g.*, CNN) such that the top-$k$ candidate points with the minimum Euclidean distance to the query point in the representation space have the same label as the query point. Note that at test time, once the representation space $h_\phi$ is learned, we can use the entire training set as the set of candidate points, akin to a standard kNN classifier. Figure 5 illustrates the proposed algorithm.

Formally, for any datapoint $\mathbf{x}$, let $\mathbf{z}$ denote a permutation of the $n$ candidate points. The uniformly-weighted kNN loss, denoted as $\ell_{\mathrm{kNN}}(\cdot)$, can be written as follows:

$$\ell_{\mathrm{kNN}}(\widehat{P}_{\mathbf{z}}, y, y_1, y_2, \ldots, y_n) = -\frac{1}{k} \sum_{j=1}^{k} \sum_{i=1}^{n} \mathbb{1}(y_i = y) \widehat{P}_{\mathbf{z}}[i, j] \tag{13}$$

Table 3: Average test kNN classification accuracies from $n$ neighbors for best value of $k$.

| Algorithm | MNIST | Fashion-MNIST | CIFAR-10 |
|---|---|---|---|
| kNN | 97.2% | 85.8% | 35.4% |
| kNN+PCA | 97.6% | 85.9% | 40.9% |
| kNN+AE | 97.6% | 87.5% | 44.2% |
| kNN + Deterministic NeuralSort | **99.5%** | **93.5%** | **90.7%** |
| kNN + Stochastic NeuralSort | 99.4% | 93.4% | 89.5% |
| CNN (w/o kNN) | 99.4% | 93.4% | **95.1%** |

where $\{y_1, y_2, \ldots, y_n\}$ are the labels for the candidate points. Note that when $\widehat{P}_{\mathbf{z}}$ is an exact permutation matrix (i.e., temperature $\tau \to 0$), this expression is exactly the negative of the fraction of $k$ nearest neighbors that have the same label as $\mathbf{x}$.

Using Eq. 13, the training objectives for Deterministic and Stochastic NeuralSort are given as:

$$\text{Deterministic:} \quad \min_\phi \frac{1}{|\mathcal{D}|} \sum_{(\mathbf{x},y) \in \mathcal{D}} \ell_{\text{kNN}}(\widehat{P}_{\text{sort}(\mathbf{s})}, y, y_1, \ldots, y_n) \tag{14}$$

$$\text{Stochastic:} \quad \min_\phi \frac{1}{|\mathcal{D}|} \sum_{(\mathbf{x},y) \in \mathcal{D}} \mathbb{E}_{\mathbf{z} \sim q(\mathbf{z}|\mathbf{s})} \left[ \ell_{\text{kNN}}(\widehat{P}_{\mathbf{z}}, y, y_1, y_2, \ldots, y_n) \right] \tag{15}$$

where each entry of $\mathbf{s}$ is given as $s_j = -\|h_\phi(\mathbf{x}) - h_\phi(\mathbf{x}_j)\|_2^2$.

**Datasets.** We consider three benchmark datasetes: MNIST dataset of handwritten digits, Fashion-MNIST dataset of fashion apparel, and the CIFAR-10 dataset of natural images (no data augmentation) with the canonical splits for training and testing.

**Baselines.** We consider kNN baselines that operate in three standard representation spaces: the canonical pixel basis, the basis specified by the top 50 principal components (PCA), an autoencoder (AE). Additionally, we experimented with $k = 1, 3, 5, 9$ nearest neighbors and across two distance metrics: uniform weighting of all $k$-nearest neighbors and weighting nearest neighbors by the inverse of their distance. For completeness, we trained a CNN with the same architecture as the one used for NeuralSort (except the final layer) using the cross-entropy loss.

**Results.** We report the classification accuracies on the standard test sets in Table 3. On both datasets, the differentiable kNN classifier outperforms all the baseline kNN variants including the convolutional autoencoder approach. The performance is much closer to the accuracy of a standard CNN.

## 7 CONCLUSION

In this paper, we proposed NeuralSort, a continuous relaxation of the sorting operator to the set of unimodal row-stochastic matrices. Our relaxation facilitates gradient estimation on any computation graph involving a `sort` operator. Further, we derived a reparameterized gradient estimator for the Plackett-Luce distribution for efficient stochastic optimization over permutations. On three illustrative tasks including a fully differentiable $k$-nearest neighbors, our proposed relaxations outperform prior work in end-to-end learning of semantic orderings of high-dimensional objects.

In the future, we would like to explore alternate relaxations to sorting as well as applications that extend widely-used algorithms such as beam search (Goyal et al., 2018). Both deterministic and stochastic NeuralSort are easy to implement. We provide reference implementations in Tensorflow (Abadi et al., 2016) and PyTorch (Paszke et al., 2017) in Appendix A. The full codebase for this work is open-sourced at `https://github.com/ermongroup/neuralsort`.

### ACKNOWLEDGEMENTS

This research was supported by NSF (#1651565, #1522054, #1733686), ONR, AFOSR (FA9550-19-1-0024), FLI, and Amazon AWS. AG is supported by MSR fellowship and Stanford Data Science scholarship. We are thankful to Jordan Alexander, Kristy Choi, Adithya Ganesh, Karan Goel, Neal Jean, Daniel Levy, Jiaming Song, Yang Song, Serena Yeung, and Hugh Zhang for feedback.

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

## APPENDICES

## A    SORTING OPERATOR

### A.1    TENSORFLOW

Sorting Relaxation for Deterministic NeuralSort:

```python
import tensorflow as tf

def deterministic_NeuralSort(s, tau):
  """
  s: input elements to be sorted. Shape: batch_size x n x 1
  tau: temperature for relaxation. Scalar.
  """

  n = tf.shape(s)[1]
  one = tf.ones((n, 1), dtype = tf.float32)

  A_s = tf.abs(s - tf.transpose(s, perm=[0, 2, 1]))
  B = tf.matmul(A_s, tf.matmul(one, tf.transpose(one)))
  scaling = tf.cast(n + 1 - 2 * (tf.range(n) + 1), dtype = tf.float32)
  C = tf.matmul(s, tf.expand_dims(scaling, 0))

  P_max = tf.transpose(C-B, perm=[0, 2, 1])
  P_hat = tf.nn.softmax(P_max / tau, -1)

  return P_hat
```

Reparameterized Sampler for Stochastic NeuralSort:

```python
def sample_gumbel(samples_shape, eps = 1e-10):

  U = tf.random_uniform(samples_shape, minval=0, maxval=1)
  return -tf.log(-tf.log(U + eps) + eps)

def stochastic_NeuralSort(s, n_samples, tau):
  """
  s: parameters of the PL distribution. Shape: batch_size x n x 1.
  n_samples: number of samples from the PL distribution. Scalar.
  tau: temperature for the relaxation. Scalar.
  """

  batch_size = tf.shape(s)[0]
  n = tf.shape(s)[1]
  log_s_perturb = s + sample_gumbel([n_samples, batch_size, n, 1])
  log_s_perturb = tf.reshape(log_s_perturb, [n_samples * batch_size, n,
      1])

  P_hat = deterministic_NeuralSort(log_s_perturb, tau)
  P_hat = tf.reshape(P_hat, [n_samples, batch_size, n, n])

  return P_hat
```

## A.2   PYTORCH

Sorting Relaxation for Deterministic NeuralSort:

```python
import torch

def deterministic_NeuralSort(s, tau):
  """
  s: input elements to be sorted. Shape: batch_size x n x 1
  tau: temperature for relaxation. Scalar.
  """

  n = s.size()[1]
  one = torch.ones((n, 1), dtype = torch.float32)

  A_s = torch.abs(s - s.permute(0, 2, 1))
  B = torch.matmul(A_s, torch.matmul(one, torch.transpose(one, 0, 1)))
  scaling = (n + 1 - 2 * (torch.arange(n) + 1)).type(torch.float32)
  C = torch.matmul(s, scaling.unsqueeze(0))

  P_max = (C-B).permute(0, 2, 1)
  sm = torch.nn.Softmax(-1)
  P_hat = sm(P_max / tau)

  return P_hat
```

Reparamterized Sampler for Stochastic NeuralSort:

```python
def sample_gumbel(samples_shape, eps = 1e-10):

    U = torch.rand(samples_shape)
    return -torch.log(-torch.log(U + eps) + eps)

def stochastic_NeuralSort(s, n_samples, tau):
  """
  s: parameters of the PL distribution. Shape: batch_size x n x 1.
  n_samples: number of samples from the PL distribution. Scalar.
  tau: temperature for the relaxation. Scalar.
  """

  batch_size = s.size()[0]
  n = s.size()[1]
  log_s_perturb = torch.log(s) + sample_gumbel([n_samples, batch_size, n,
      1])
  log_s_perturb = log_s_perturb.view(n_samples * batch_size, n, 1)

  P_hat = deterministic_NeuralSort(log_s_perturb, tau)
  P_hat = P_hat.view(n_samples, batch_size, n, n)

  return P_hat
```

## B   PROOFS OF THEORETICAL RESULTS

### B.1   LEMMA 2

*Proof.* For any value of $\lambda$, the following inequalities hold:

$$\sum_{i=1}^{k} s_{[i]} = \lambda k + \sum_{i=1}^{k}(s_{[i]} - \lambda)$$

$$\leq \lambda k + \sum_{i=1}^{k} \max(s_{[i]} - \lambda, 0)$$

$$\leq \lambda k + \sum_{i=1}^{n} \max(s_i - \lambda, 0).$$

Furthermore, for $\lambda = s_{[k]}$:

$$\lambda k + \sum_{i=1}^{n} \max(s_i - \lambda, 0) = s_{[k]}k + \sum_{i=1}^{n} \max(s_i - s_{[k]}, 0)$$

$$= s_{[k]}k + \sum_{i=1}^{k}(s_{[i]} - s_{[k]})$$

$$= \sum_{i=1}^{k} s_{[i]}.$$

This finishes the proof. $\qquad\square$

### B.2   COROLLARY 3

*Proof.* We first consider at exactly what values of $\lambda$ the sum in Lemma 2 is minimized. For simplicity we will only prove the case where all values of $\mathbf{s}$ are distinct.

The equality $\sum_{i=1}^{k} s_{[i]} = \lambda k + \sum_{i=1}^{n} \max(s_i - \lambda, 0)$ holds only when $s_{[k]} \leq \lambda \leq s_{[k+1]}$. By Lemma 2, these values of $\lambda$ also minimize the RHS of the equality.

Symmetrically, if one considers the score vector $\mathbf{t} = -\mathbf{s}$, then $\lambda(n - k + 1) + \sum_{i=1}^{n} \max(t_i - \lambda, 0)$ is minimized at $t_{[n-k+1]} \leq \lambda \leq t_{[n-k+2]}$.

Replacing $\lambda$ by $-\lambda$ and using the definition of $\mathbf{t}$ implies that $\lambda(k - 1 - n) + \sum_{i=1}^{n} \max(\lambda - s_i, 0)$ is minimized at $s_{[k-1]} \leq \lambda \leq s_{[k]}$.

It follows that:

$$s_{[k]} = \arg\min_{\lambda \in \mathbf{s}} \left( \lambda k + \sum_{i=1}^{n} \max(s_i - \lambda, 0) \right) + \left( \lambda(k - 1 - n) + \sum_{i=1}^{n} \max(\lambda - s_i, 0) \right)$$

$$= \arg\min_{\lambda \in \mathbf{s}} \lambda(2k - 1 - n) + \sum_{i=1}^{n} |s_i - \lambda|.$$

Thus, if $s_i = s_{[k]}$, then $i = \arg\min(2k - 1 - n)\mathbf{s} + A_{\mathbf{s}}\mathbb{1}$. This finishes the proof.

$\qquad\square$

### B.3   THEOREM 4

We prove the two properties in the statement of the theorem independently:

1. *Unimodality*

   *Proof.* By definition of the softmax function, the entries ef $\widehat{P}$ are positive and sum to 1. To show that $\widehat{P}$ satisfies the argmax permutation property, . Formally, for any given row $i$, we construct the argmax permutation vector $\mathbf{u}$ as:

   $$u_i = \arg\max[\text{soft}\max((n+1-2i)\mathbf{s} - A_\mathbf{s}\mathbb{1})]$$
   $$= \arg\max[(n+1-2i)\mathbf{s} - A_\mathbf{s}\mathbb{1}]$$
   $$= [i]$$

   where the square notation $[i]$ denotes the index of the $i$-th largest element. The first step follows from the fact that the softmax function is monotonically increasing and hence, it preserves the argmax. The second equality directly follows from Corollary 3. By definition, $\texttt{sort}(\mathbf{s}) = \{[1], [2], \ldots, [n]\}$, finishing the proof. $\qquad\square$

2. *Limiting behavior*

   *Proof.* As shown in Gao & Pavel (2017), the softmax function may be equivalently defined as $\text{soft}\max(z/\tau) = \arg\max_{x\in\Delta^{n-1}}\langle x, z\rangle - \tau\sum_{i=1}^n x_i\log x_i$. In particular, $\lim_{\tau\to 0}\text{soft}\max(z/\tau) = \arg\max x$. The distributional assumptions ensure that the elements of $\mathbf{s}$ are distinct a.s., so plugging in $z = (n+1-2k)\mathbf{s} - A_\mathbf{s}\mathbb{1}$ completes the proof. $\qquad\square$

## B.4 PROPOSITION 5

This result follows from an earlier result by Yellott Jr (1977). We give the proof sketch below and refer the reader to Yellott Jr (1977) for more details.

*Sketch.* Consider random variables $\{X_i\}_{i=1}^n$ such that $X_i \sim \text{Exp}(s_{z_i})$.

We may prove by induction a generalization of the memoryless property:

$$q(X_1 \leq \cdots \leq X_n | x \leq \min_i X_i)$$
$$= \int_0^\infty q(x \leq X_1 \leq x+t | x \leq \min_i X_i)q(X_2 \leq \cdots \leq X_n | x+t \leq \min_{i\geq 2} X_i)\mathrm{d}t$$
$$= \int_0^\infty q(0 \leq X_1 \leq t)q(X_2 \leq \cdots \leq X_n | x+t \leq \min_{i\geq 2} X_i)\mathrm{d}t.$$

If we assume as inductive hypothesis that $q(X_2 \leq \cdots \leq X_n | x+t \leq \min_{i\geq 2} X_i) = q(X_2 \leq \cdots \leq X_n | t \leq \min_{i\geq 2} X_i)$, we complete the induction as:

$$q(X_1 \leq \cdots \leq X_n | x \leq \min_i X_i)$$
$$= \int_0^\infty q(0 \leq X_1 \leq t)q(X_2 \leq \cdots \leq X_n | t \leq \min_{i\geq 2} X_i)\mathrm{d}t$$
$$= q(X_1 \leq X_2 \leq \cdots \leq X_n | 0 \leq \min_i X_i).$$

It follows from a familiar property of argmin of exponential distributions that:

$$q(X_1 \leq X_2 \leq \cdots \leq X_n) = q(X_1 \leq \min_i X_i)q(X_2 \leq \cdots \leq X_n | X_1 \leq \min_i X_i)$$
$$= \frac{s_{z_1}}{Z}q(X_2 \leq \cdots \leq X_n | X_1 \leq \min_i X_i)$$
$$= \frac{s_{z_1}}{Z}\int_0^\infty q(X_1 = x)q(X_2 \leq \cdots \leq X_n | x \leq \min_{i\geq 2} X_i)\mathrm{d}x$$
$$= \frac{s_{z_1}}{Z}q(X_2 \leq \cdots \leq X_n),$$

and by another induction, we have $q(X_1 \leq \cdots \leq X_n) = \prod_{i=1}^{n} \frac{s_{z_i}}{Z - \sum_{k=1}^{i-1} s_{z_k}}$.

Finally, following the argument of Balog et al. (2017), we apply the strictly decreasing function $g(x) = -\beta \log x$ to this identity, which from the definition of the Gumbel distribution implies:

$$q(\tilde{s}_{z_1} \geq \cdots \geq \tilde{s}_{z_n}) = \prod_{i=1}^{n} \frac{s_{z_i}}{Z - \sum_{k=1}^{i-1} s_{z_k}}.$$

$\square$

## C    ARG MAX SEMANTICS FOR TIED MAX ELEMENTS

While applying the $\arg\max$ operator to a vector with duplicate entries attaining the $\max$ value, we need to define the operator semantics for $\arg\max$ to handle ties in the context of the proposed relaxation.

**Definition 6.** *For any vector with ties, let* $\arg\max$ set *denote the operator that returns the set of all indices containing the* $\max$ *element. We define the* $\arg\max$ *of the $i$-th in a matrix $M$ recursively:*

1. *If there exists an index $j \in \{1, 2, \dots, n\}$ that is a member of $\arg\max$ set$(M[i,:])$ and has not been assigned as an $\arg\max$ of any row $k < i$, then the $\arg\max$ is the smallest such index.*

2. *Otherwise, the $\arg\max$ is the smallest index that is a member of the $\arg\max$ set$(M[i,:])$.*

This function is efficiently computable with additional bookkeeping.

**Lemma 7.** *For an input vector $\mathbf{s}$ with the sort permutation matrix given as $P_{sort(\mathbf{s})}$, we have $s_{j_1} = s_{j_2}$ if and only if there exists a row $i$ such that $\widehat{P}[i, j_1] = \widehat{P}[i, j_2]$ for all $j_1, j_2 \in \{1, 2, \dots, n\}$.*

*Proof.* From Eq. 5, we have the $i$-th row of $\widehat{P}[i,:]$ given as:

$$\widehat{P}[i,:] = \text{soft}\max\left[((n+1-2i)\mathbf{s} - A_{\mathbf{s}}\mathbb{1})/\tau\right]$$

. Therefore, we have the equations:

$$\widehat{P}[i, j_1] = \frac{\exp(((n+1-2i)s_{j_1} - (A_{\mathbf{s}}\mathbb{1})_i)/\tau)}{Z}$$

$$= \widehat{P}[i, j_2] = \frac{\exp(((n+1-2i)s_{j_2} - (A_{\mathbf{s}}\mathbb{1})_i)/\tau)}{Z}$$

for some fixed normalization constant $Z$. As the function $f(x) = \frac{\exp(((n+1-2i)x - (A_{\mathbf{s}}\mathbb{1})_i)/\tau)}{Z}$ is invertible, both directions of the lemma follow immediately. $\square$

**Lemma 8.** *If* $\arg\max$ set$(\widehat{P}[i_1,:])$ *and* $\arg\max$ set$(\widehat{P}[i_2,:])$ *have a non-zero intersection, then* $\arg\max$ set$(\widehat{P}[i_1,:]) = \arg\max$ set$(\widehat{P}[i_2,:])$.

*Proof.* Assume without loss of generality that $|\arg\max$ set$(\widehat{P}[i_1,:])| > 1$ for some $i$. Let $j_1, j_2$ be two members of $|\arg\max$ set$(\widehat{P}[i_1,:])|$. By Lemma 7, $s_{j_1} = s_{j_2}$, and therefore $\widehat{P}[i_2, j_1] = \widehat{P}[i_2, j_2]$. Hence if $j_1 \in \arg\max$ set$(\widehat{P}[i_2,:])$, then $j_2$ is also an element. A symmetric argument implies that if $j_2 \in \arg\max$ set$(\widehat{P}[i_2,:])$, then $j_1$ is also an element for arbitrary $j_1, j_2 \in |\arg\max$ set$(\widehat{P}[i_1,:])|$. This completes the proof. $\square$

**Proposition 9.** *(Argmax Permutation with Ties) For $\mathbf{s} = [s_1, s_2, \dots, s_n]^T \in \mathbb{R}^n$, the vector $\mathbf{z}$ defined by $z_i = \arg\max_j \widehat{P}_{sort(\mathbf{s})}[i, j]$ is such that $\mathbf{z} \in \mathcal{Z}_n$.*

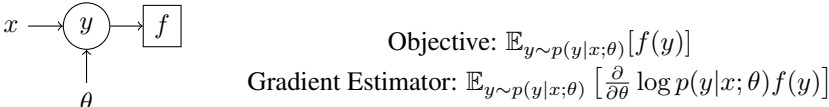

$$\text{Objective: } \mathbb{E}_{y \sim p(y|x;\theta)}[f(y)]$$

$$\text{Gradient Estimator: } \mathbb{E}_{y \sim p(y|x;\theta)}\left[\frac{\partial}{\partial \theta} \log p(y|x;\theta) f(y)\right]$$

Figure 6: A stochastic computation graph for an arbitrary input $x$, intermediate node $y$, and a single parameter $\theta$. Squares denote deterministic nodes and circles denote stochastic nodes.

*Proof.* From Corollary 3, we know that the row $\widehat{P}_{\texttt{sort}(\mathbf{s})}[i,:]$ attains its maximum (perhaps non-uniquely) at some $\widehat{P}_{\texttt{sort}(\mathbf{s})}[i,j]$ where $s_j = s_{[i]}$. Note that $s_{[i]}$ is well-defined even in the case of ties.

Consider an arbitrary row $\widehat{P}_{\texttt{sort}(\mathbf{s})}[i,:]$ and let $\arg\max\text{set}(\widehat{P}_{\texttt{sort}(\mathbf{s})}[i,:]) = \{j_1, \ldots, j_m\}$. It follows from Lemma 7 that there are exactly $m$ scores in $\mathbf{s}$ that are equal to $s_{[i]}$: $s_{j_1}, \ldots, s_{j_m}$. These scores corresponds to $m$ values of $s_{[i']}$ such that $s_{[i']} = s_{[i]}$, and consequently to $m$ rows $P[i',:]$ that are maximized with values $s_{[i']} = s_{[i]}$ and consequently (by Lemma 8) at indices $j_1, \ldots, j_m$.

Suppose we now chose an $i'$ such that $s_{[i']} \neq s_{[i]}$. Then $\widehat{P}[i',:]$ attains its maximum at some $\widehat{P}\texttt{sort}(\mathbf{s})[i',j']$ where $s_{j'} = s_{[i']}$. Because $s_{j'} = s_{[i']} \neq s_{[i]} = s_j$, Lemma 7 tells us that $\widehat{P}[i',:]$ does not attain its maximum at any of $j_1, \ldots, j_m$. Therefore, only $m$ rows have a non-zero $\arg\max$ set intersection with $\arg\max\text{set}(P[i,:])$.

Because $P[i,:]$ is one of these rows, there can be up to $m-1$ such rows above it. Because each row above only has one $\arg\max$ assigned via the tie-breaking protocol, it is only possible for up to $m-1$ elements of $\arg\max\text{set}(P[i,:])$ to have been an $\arg\max$ of a previous row $k < i$. As $|\arg\max\text{set}(P[i,:])| = m$, there exists at least one element that has not been specified as the $\arg\max$ of a previous row (pigeon-hole principle). Thus, the $\arg\max$ of each row are distinct. Because each argmax is also an element of $\{1, \ldots, n\}$, it follows that $\mathbf{z} \in \mathscr{Z}_n$. $\qquad\square$

# D   EXPERIMENTAL DETAILS AND ANALYSIS

We used Tensorflow (Abadi et al., 2016) and PyTorch (Paszke et al., 2017) for our experiments. In Appendix A, we provide "plug-in" snippets for implementing our proposed relaxations in both Tensorflow and PyTorch. The full codebase for reproducing the experiments can be found at `https://github.com/ermongroup/neuralsort`.

For the sorting and quantile regression experiments, we used standard training/validation/test splits of $50,000/10,000/10,000$ images of MNIST for constructing the large-MNIST dataset. We ensure that only digits in the standard training/validation/test sets of the MNIST dataset are composed together to generate the corresponding sets of the large-MNIST dataset. For CIFAR-10, we used a split of $45,000/5000/10,000$ examples for training/validation/test. With regards to the baselines considered, we note that the REINFORCE based estimators were empirically observed to be worse than almost all baselines for all our experiments.

## D.1   SORTING HANDWRITTEN NUMBERS

**Architectures.**   We control for the choice of computer vision models by using the same convolutional network architecture for each sorting method. This architecture is as follows:

Conv[Kernel: 5x5, Stride: 1, Output: 140x28x32, Activation: Relu]

$\rightarrow$ Pool[Stride: 2, Output: 70x14x32]

$\rightarrow$ Conv[Kernel: 5x5, Stride: 1, Output: 70x14x64, Activation: Relu]

$\rightarrow$ Pool[Stride: 2, Output: 35x7x64]

$\rightarrow$ FC[Units: 64, Activation: Relu]

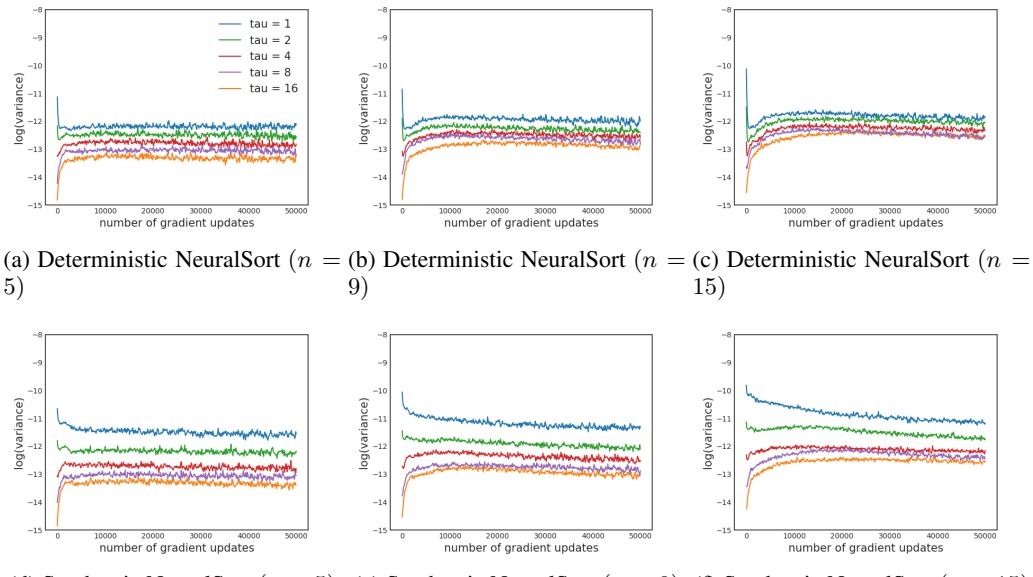

(a) Deterministic NeuralSort ($n = 5$)  (b) Deterministic NeuralSort ($n = 9$)  (c) Deterministic NeuralSort ($n = 15$)

(d) Stochastic NeuralSort ($n = 5$)  (e) Stochastic NeuralSort ($n = 9$)  (f) Stochastic NeuralSort ($n = 15$)

Figure 7: Running average of the log-variance in gradient estimates during training for varying temperatures $\tau$.

Note that the dimension of a 5-digit large-MNIST image is $140 \times 28$. The primary difference between our methods is how we combine the scores to output a row-stochastic prediction matrix.

For NeuralSort-based methods, we use another fully-connected layer of dimension 1 to map the image representations to $n$ scalar scores. In the case of Stochastic NeuralSort, we then sample from the PL distribution by perturbing the scores multiple times with Gumbel noise. Finally, we use the NeuralSort operator to map the set of $n$ scores (or each set of $n$ perturbed scores) to its corresponding unimodal row-stochastic matrix.

For Sinkhorn-based methods, we use a fully-connected layer of dimension $n$ to map each image to an $n$-dimensional vector. These vectors are then stacked into an $n \times n$ matrix. We then either map this matrix to a corresponding doubly-stochastic matrix (Sinkhorn) or sample directly from a distribution over permutation matrices via Gumbel perturbations (Gumbel-Sinkhorn). We implemented the Sinkhorn operator based on code snippets obtained from the open source implementation of Mena et al. (2018) available at `https://github.com/google/gumbel_sinkhorn`.

For the Vanilla RS baseline, we ran each element through a fully-connected $n$ dimensional layer, concatenated the representations of each element and then fed the results through three fully-connected $n^2$-unit layers to output multiclass predictions for each rank.

All our methods yield row-stochastic $n \times n$ matrices as their final output. Our loss is the row-wise cross-entropy loss between the true permutation matrix and the row-stochastic output.

**Hyperparameters.** For this experiment, we used an Adam optimizer with an initial learning rate of $10^{-4}$ and a batch size of 20. Continuous relaxations to sorting also introduce another hyperparameter: the temperature $\tau$ for the Sinkhorn-based and NeuralSort-based approaches. We tuned this hyperparameter on the set $\{1, 2, 4, 8, 16\}$ by picking the model with the best validation accuracy on predicting entire permutations (as opposed to predicting individual maps between elements and ranks).

**Effect of temperature.** In Figure 7, we report the log-variance in gradient estimates as a function of the temperature $\tau$. Similar to the effect of temperature observed for other continuous relaxations to discrete objects such as Gumbel-softmax (Jang et al., 2017; Maddison et al., 2017), we note that higher temperatures lead to lower variance in gradient estimates. The element-wise mean squared

Table 4: Element-wise mean squared difference between unimodal approximations and the projected hard permutation matrices for the best temperature $\tau$, averaged over the test set.

| Algorithm | $n = 3$ | $n = 5$ | $n = 7$ | $n = 9$ | $n = 15$ |
|---|---|---|---|---|---|
| Deterministic NeuralSort | 0.0052 | 0.0272 | 0.0339 | 0.0105 | 0.0220 |
| Stochastic NeuralSort | 0.0095 | 0.0327 | 0.0189 | 0.0111 | 0.0179 |

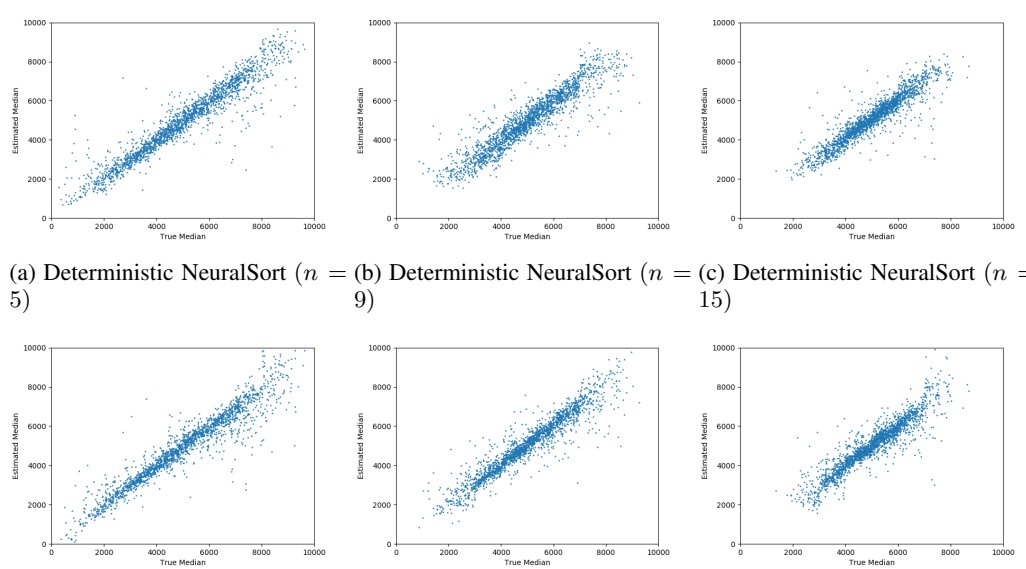

(a) Deterministic NeuralSort ($n =$ 5)   (b) Deterministic NeuralSort ($n =$ 9)   (c) Deterministic NeuralSort ($n =$ 15)

(d) Stochastic NeuralSort ($n = 5$)   (e) Stochastic NeuralSort ($n = 9$)   (f) Stochastic NeuralSort ($n = 15$)

Figure 8: True vs. predicted medians for quantile regression on the large-MNIST dataset.

difference between unimodal approximations $\widehat{P}_{\texttt{sort}(\mathbf{s})}$ and the projected hard permutation matrices $P_{\texttt{sort}(\mathbf{s})}$ for the best $\tau$ on the test set is shown in Table 4.

## D.2 QUANTILE REGRESSION

**Architectures.** Due to resource constraints, we ran the quantile regression experiment on 4-digit numbers instead of 5-digit numbers. We use the same neural network architecture as previously used in the sorting experiment.

Conv[Kernel: 5x5, Stride: 1, Output: 112x28x32, Activation: Relu]
$\rightarrow$ Pool[Stride: 2, Output: 56x14x32]
$\rightarrow$ Conv[Kernel: 5x5, Stride: 1, Output: 56x14x64, Activation: Relu]
$\rightarrow$ Pool[Stride: 2, Output: 28x7x64]
$\rightarrow$ FC[Units: 64, Activation: Relu]

The vanilla NN baseline for quantile regression was generated by feeding the CNN representations into a series of three fully-connected layers of ten units each, the last of which mapped to a single-unit estimate of the median. In the other experiments, one copy of this network was used to estimate each element's rank through a method like Gumbel-Sinkhorn or NeuralSort that produces a row-stochastic matrix, while another copy was used to estimate each element's value directly. Point predictions are obtained by multiplying the center row of the matrix with the column vector of estimated values, and we minimize the $\ell_2$ loss between these point predictions and the true median, learning information about ordering and value simultaneously.

**Hyperparameters.** We used the Adam optimizer with an initial learning rate of $10^{-4}$ and a batch size of 5. The temperature $\tau$ was tuned on the set $\{1, 2, 4, 8, 16\}$ based on the validation loss.

**Further Analysis.** In Figure 8, we show the scatter plots for the true vs. predicted medians on 2000 test points from the large-MNIST dataset as we vary $n$. For stochastic NeuralSort, we average the predictions across 5 samples. As we increase $n$, the distribution of true medians concentrates, leading to an easier prediction problem (at an absolute scale) and hence, we observe lower MSE for larger $n$ in Table 2. However, the relatively difficulty of the problem increases with increasing $n$, as the model is trying to learn a semantic sorting across a larger set of elements. This is reflected in the $R^2$ values in Table 2 which show a slight dip as $n$ increases.

### D.3 END-TO-END, DIFFERENTIABLE $k$-NEAREST NEIGHBORS

**Architectures.** The baseline kNN implementation for the pixel basis, PCA basis and the autoencoder basis was done using `sklearn`. For the autoencoder baselines for kNN, we used the following standard architectures.

**MNIST and Fashion-MNIST:** The dimension of the encoding used for distance computation in kNN is 50.

FC[Units: 500, Activation: Relu]
$\rightarrow$ FC[Units: 500, Activation: Relu]
$\rightarrow$ FC[Units: 50, Activation: Relu]
$=$ *(embedding)*
$\rightarrow$ FC[Units: 500, Activation: Relu]
$\rightarrow$ FC[Units: 500, Activation: Relu]
$\rightarrow$ FC[Units: 784, Activation: Sigmoid]

**CIFAR-10:** The dimension of the encoding used for distance computation in kNN is 256. The architecture and training procedure follows the one available at `https://github.com/shibuiwilliam/Keras_Autoencoder`.

Conv[Kernel: 3x3, Stride: 1, Output: 32x32x64, Activation: Relu]
$\rightarrow$ Pool[Stride: 2, Output: 16x16x64]
$\rightarrow$ Conv[Kernel: 3x3, Stride: 1, Output: 16x16x32, Normalization: BatchNorm, Activation: Relu]
$\rightarrow$ Pool[Stride: 2, Output: 8x8x32]
$\rightarrow$ Conv[Stride: 3, Output: 8x8x16, Normalization: BatchNorm, Activation: Relu]
$\rightarrow$ MaxPool[Stride: 2, Output: 4x4x16]
$=$ *(embedding)*
$\rightarrow$ Conv[Kernel: 3x3, Stride: 1, Output: 4x4x16, Normalization: BatchNorm, Activation: Relu]
$\rightarrow$ UpSampling[Size: 2x2, Output: 8x8x16]
$\rightarrow$ Conv[Kernel: 3x3, Stride: 1, Output: 8x8x32, Normalization: BatchNorm, Activation: Relu]
$\rightarrow$ UpSampling[Size: 2x2, Output: 16x16x32]
$\rightarrow$ Conv[Kernel: 3x3, Output: 16x16x64, Normalization: BatchNorm, Activation: Relu]
$\rightarrow$ UpSampling[Size: 2x2, Output: 32x32x64]
$\rightarrow$ Conv[Kernel: 3x3, Stride: 1, Output: 32x32x3, Normalization: BatchNorm, Activation: Sigmoid]

Table 5: Accuracies of Deterministic and Stochastic NeuralSort for differentiable $k$-nearest neighbors, broken down by $k$.

| Dataset | $k$ | Deterministic NeuralSort | Stochastic NeuralSort |
|---------|-----|--------------------------|------------------------|
| MNIST | 1 | 99.2% | 99.1% |
| | 3 | **99.5%** | 99.3% |
| | 5 | 99.3% | **99.4%** |
| | 9 | 99.3% | **99.4%** |
| Fashion-MNIST | 1 | 92.6% | 92.2% |
| | 3 | 93.2% | 93.1% |
| | 5 | **93.5%** | 93.3% |
| | 9 | 93.0% | **93.4%** |
| CIFAR-10 | 1 | 88.7% | 85.1% |
| | 3 | 90.0% | 87.8% |
| | 5 | 90.2% | 88.0% |
| | 9 | **90.7%** | **89.5%** |

For the MNIST experiments with NeuralSort, we used a network similar to the large-MNIST network used in the previous experiments:

> Conv[Kernel: 5x5, Stride: 1, Output: 24x24x20, Activation: Relu]
> $\rightarrow$ Pool[Stride: 2, Output: 12x12x20]
> $\rightarrow$ Conv[Kernel: 5x5, Stride: 1, Output: 8x8x50, Activation: Relu]
> $\rightarrow$ Pool[Stride: 2, Output: 4x4x50]
> $\rightarrow$ FC[Units: 500, Activation: Relu]

For the Fashion-MNIST and CIFAR experiments with NeuralSort, we use the ResNet18 architecture as described in `https://github.com/kuangliu/pytorch-cifar`.

**Hyperparameters.** For this experiment, we used an SGD optimizer with a momentum parameter of 0.9, with a batch size of 100 queries and 100 neighbor candidates at a time. We chose the temperature hyperparameter from the set $\{1, 16, 64\}$, the constant learning rate from $\{10^{-4}, 10^{-5}\}$, and the number of nearest neighbors $k$ from the set $\{1, 3, 5, 9\}$. The model with the best evaluation loss was evaluated on the test set. We suspect that accuracy improvements can be made by a more expensive hyperparameter search and a more fine-grained learning rate schedule.

**Accuracy for different $k$.** In Table 5, we show the performance of Deterministic and Stochastic NeuralSort for different choice of the hyperparameter $k$ for the differentiable $k$-nearest neighbors algorithm.

# E  LOSS FUNCTIONS

For each of the experiments in this work, we assume we have access to a finite dataset $\mathcal{D} = \{(\mathbf{x}^{(1)}, \mathbf{y}^{(1)}), (\mathbf{x}^{(2)}, \mathbf{y}^{(2)}), \ldots\}$. Our goal is to learn a predictor for $\mathbf{y}$ given $\mathbf{x}$, as in a standard supervised learning (classification/regression) setting. Below, we state and elucidate the semantics of the training objective optimized by Deterministic and Stochastic NeuralSort for the sorting and quantile regression experiments.

## E.1  SORTING HANDWRITTEN NUMBERS

We are given a dataset $\mathcal{D}$ of sequences of large-MNIST images and the permutations that sort the sequences. That is, every datapoint in $\mathcal{D}$ consists of an input $\mathbf{x}$, which corresponds to a sequence containing $n$ images, and the desired output label $\mathbf{y}$, which corresponds to the permutation that sorts this sequence (as per the numerical values of the images in the input sequence). For example, Figure 4 shows one input sequence of $n = 5$ images, and the permutation $\mathbf{y} = [3, 5, 1, 4, 2]$ that sorts this sequence.

For any datapoint $\mathbf{x}$, let $\ell_{\mathrm{CE}}(\cdot)$ denote the average multiclass cross entropy (CE) error between the rows of the true permutation matrix $P_{\mathbf{y}}$ and a permutation matrix $P_{\widehat{\mathbf{y}}}$ corresponding to a predicted permutation, say $\widehat{\mathbf{y}}$.

$$\ell_{\mathrm{CE}}(P_{\mathbf{y}}, P_{\widehat{\mathbf{y}}}) = \frac{1}{n} \sum_{i=1}^{n} \sum_{j=1}^{n} \mathbb{1}(P_{\mathbf{y}}[i,j] = 1) \log P_{\widehat{\mathbf{y}}}[i,j]$$

where $\mathbb{1}(\cdot)$ denotes the indicator function. Now, we state the training objective functions for the Deterministic and Stochastic NeuralSort approaches respectively.

1. Deterministic NeuralSort

$$\min_{\phi} \frac{1}{|\mathcal{D}|} \sum_{(\mathbf{x}, \mathbf{y}) \in \mathcal{D}} \ell_{\mathrm{CE}}(P_{\mathbf{y}}, \widehat{P}_{\mathrm{sort}(\mathbf{s})}) \tag{16}$$

where each entry of $\mathbf{s}$ is given as $s_j = h_{\phi}(\mathbf{x}_j)$.

2. Stochastic NeuralSort

$$\min_{\phi} \frac{1}{|\mathcal{D}|} \sum_{(\mathbf{x}, \mathbf{y}) \in \mathcal{D}} \mathbb{E}_{\mathbf{z} \sim q(\mathbf{z}|\mathbf{s})} \left[ \ell_{\mathrm{CE}}(P_{\mathbf{y}}, \widehat{P}_{\mathbf{z}})) \right] \tag{17}$$

where each entry of $\mathbf{s}$ is given as $s_j = h_{\phi}(\mathbf{x}_j)$.

To ground this in our experimental setup, the score $s_j$ for each large-MNIST image $\mathbf{x}_j$ in any input sequence $\mathbf{x}$ of $n = 5$ images is obtained via a CNN $h_{\phi}()$ with parameters $\phi$. Note that the CNN parameters $\phi$ are shared across the different images $\mathbf{x}_1, \mathbf{x}_2, \ldots, \mathbf{x}_n$ in the sequence for efficient learning.

### E.2 QUANTILE REGRESSION

In contrast to the previous experiment, here we are given a dataset $\mathcal{D}$ of sequences of large-MNIST images and only the numerical value of the median element for each sequence. For example, the desired label corresponds to $y = 2960$ (a real-valued scalar) for the input sequence of $n = 5$ images in Figure 4.

For any datapoint $\mathbf{x}$, let $\ell_{\mathrm{MSE}}(\cdot)$ denote the mean-squared error between the true median $y$ and the prediction, say $\widehat{y}$.

$$\ell_{\mathrm{MSE}}(y, \widehat{y}) = \|y - \widehat{y}\|_2^2$$

For the NeuralSort approaches, we optimize the following objective functions.

1. Deterministic NeuralSort

$$\min_{\phi, \theta} \frac{1}{|\mathcal{D}|} \sum_{(\mathbf{x}, y) \in \mathcal{D}} \ell_{\mathrm{MSE}}(y, g_{\theta}(\widehat{P}_{\mathrm{sort}(\mathbf{s})}\mathbf{x})) \tag{18}$$

where each entry of $\mathbf{s}$ is given as $s_j = h_{\phi}(\mathbf{x}_j)$.

2. Stochastic NeuralSort

$$\min_{\phi, \theta} \frac{1}{|\mathcal{D}|} \sum_{(\mathbf{x}, \mathbf{y}) \in \mathcal{D}} \mathbb{E}_{\mathbf{z} \sim q(\mathbf{z}|\mathbf{s})} \left[ \ell_{\mathrm{MSE}}(y, g_{\theta}(\widehat{P}_{\mathbf{z}}\mathbf{x})) \right] \tag{19}$$

where each entry of $\mathbf{s}$ is given as $s_j = h_{\phi}(\mathbf{x}_j)$.

As before, the score $s_j$ for each large-MNIST image $\mathbf{x}_j$ in any input sequence $\mathbf{x}$ of $n$ images is obtained via a CNN $h_{\phi}()$ with parameters $\phi$. Once we have a predicted permutation matrix $\widehat{P}_{\mathrm{sort}(\mathbf{s})}$ (or $\widehat{P}_{\mathbf{z}}$) for deterministic (or stochastic) approaches, we extract the median image via $\widehat{P}_{\mathrm{sort}(\mathbf{s})}\mathbf{x}$ (or $\widehat{P}_{\mathbf{z}}\mathbf{x}$). Finally, we use a neural network $g_{\theta}(\cdot)$ with parameters $\theta$ to regress this image to a scalar prediction for the median.

