# OpenReview forum: "Stochastic Optimization of Sorting Networks via Continuous Relaxations"
_ICLR.cc/2019/Conference_

### Official Review · AnonReviewer2 · 2018-10-27
**Interesting theoretical results, but connection to the experimental results is not clear**

**Rating:** 6
**Confidence:** 3

**Review:**

In many machine learning applications, sorting is an important step such as ranking. However, the sorting operator is not differentiable with respect to its inputs. The main idea of the paper is to introduce a continuous relaxation of the sorting operator in order to construct an end-to-end gradient-based optimization. This relaxation is introduced as \hat{P}_{sort(s)} (see Equation 4). The paper also introduces a stochastic extension of its method
using Placket-Luce distributions and Monte Carlo. Finally, the introduced deterministic and stochastic methods are evaluated experimentally in 3 different applications: 1. sorting handwritten numbers, 2. Quantile regression, and 3. End-to-end differentiable k-Nearest Neighbors.

The introduction of the differentiable approximation of the sorting operator is interesting and seems novel. However, the paper is not well-written and it is hard to follow the paper especially form Section 4 and on. It is not clear how the theoretical results in Section 3 and 4 are used for the experiments in Section 6. For instance:
** In page 4, what is "s" in the machine learning application?
** In page 4, in Equation 6, what are theta, s, L and f exactly in our machine learning applications?

Remark:
** The phrase "Sorting Networks" in the title of the paper is confusing. This term typically refers to a network of comparators applied to a set of N wires (See e.g. [1])
** Page 2 -- Section 2 PRELIMINARIES -- It seems that sort(s) must be [1,4,2,3].

[1] Ajtai M, Komlós J, Szemerédi E. An 0 (n log n) sorting network. InProceedings of the fifteenth annual ACM symposium on Theory of computing 1983 Dec 1 (pp. 1-9). ACM

---

> ### Author Response · Authors · 2018-11-16
> **Response to reviewer questions and feedback**
>
> Thanks for reviewing our paper and the helpful feedback! We have addressed your questions and comments below.
>
> Q1. Clarity in Sections 3, 4. Connection with experiments.
> A1. Following up on the reviewer’s feedback, we have made the following edits in the revised version:
> - Edited and expanded the introductory paragraphs for Section 3 and Section 4 to ensure a smooth transition.
> - Revised Figures 4, 5 (which were previously Figure 3, 4 in the in the first version of the paper) to clearly indicate the scores “s” for each experiment.
> - Included a new Appendix E which formally states the loss functions optimized by the Sortnet approaches for all three experiments.
>
> For the specific follow-up questions in the review, we first note that Equation (7) (which was previously Equation (6) in the first version of the paper) is the general style of expressions used in the relevant literature on stochastic optimization, see e.g., Section 3 in Jang et al., 2017. These expressions are succinct, but as the reviewer points out, they need additional clarification when extended to the experiments. We hope Appendix E will help clarify these formally. For completeness, we address the two questions specifically raised by the reviewer here:
>
> In all our experiments, we are dealing with sequences of n objects x = [x1, x2, …, xn] and trying to sort these objects for an end goal. In Section 6.1, the goal is to output the sorted permutation for a sequence of n largeMNIST images; in 6.2, the goal is to output the median value in the sequence; in 6.3, the goal is to sort a sequence of training points as per their distances to a query point for kNN classification. We now explain the notation in the context of largeMNIST experiments in Section 6.1/6.2 which share the same experimental setup and dataset; the kNN experiments in Section 6.3 follow similarly.
>
> - s=[s1, s2, …, sn] corresponds to a vector of scores, one for each largeMNIST image in the input sequence. Each score si is the output of a CNN which takes as input an image xi. The CNNs across the different largeMNIST images x1, x2, ..., xn share parameters. Note that we directly specify the vector s (and skip x as well as the CNN parameters relating x to s) in Equation (7) for brevity. In Section 4, we derived gradients of the objective w.r.t. s, which can be backpropagated via chain rule to update the CNN parameters in a straightforward manner.
> - q is the Plackett-Luce distribution over permutations z and parameterized by scores s.
> - f is any function (that optionally depends on additional parameters \theta) that acts over a permutation matrix P_z. In the experiments in Section 6.1, the function f is the element-wise cross-entropy loss between the true permutation matrix that sorts x and P_z. Again for the purpose of generality , we do not explicitly include the ground-truth permutation as an argument to the function f in Equation (7) since such objectives also arise in unsupervised settings, e.g., latent variable modeling where there is no ground-truth label.
> - The parameters \theta for specifying f as a function of P_z are optional and task-specific. In particular, the cross-entropy loss function f for experiments in Section 6.1 does not needs any additional parameters \theta. For the experiments in Section 6.2, we cannot compute a loss directly with respect to the permutation matrix P_z since we need to regress a scalar value for the median. Instead, we feed the predicted median image in the input sequence (can be obtained via sorting x as per P_z) to a neural network (with parameters \theta) to obtain a real-valued, scalar prediction. We then compute f as the the MSE regression loss between the true median value and the value predicted by the parameterized neural network.
> - Lastly, L denotes the expected value of the objective function f w.r.t. the distribution q.
>
> Please refer to Figures 4, 5 for the computational pipeline and Appendix E for the precise loss functions for each experiment. Let us know if there is any other detail that needs clarification!
>
> Q2. Confusing use of the phrase "Sorting Networks" in the title of the paper.
> A2. Thanks for pointing it out! If permitted by the conference rules, we will consider substituting ‘networks’ to ‘operators’ in the title of the final version of the paper.
>
> Q3. Page 2 -- Section 2 PRELIMINARIES -- It seems that sort(s) must be [1,4,2,3].
> A3. We believe the sort(s) expression in the paper is correct. This is because the largest element (=9) is at index 1, second largest element (=5) is at index 3, third largest element (=2) is at index 4 and the smallest element (=1) is at index 2. Hence, sort(s)=[1,3,4,2]^T as indicated in the paper.

---

### Official Review · AnonReviewer3 · 2018-11-02
**Nice results**

**Rating:** 7
**Confidence:** 3

**Review:**

After responses: I now understand the paper, and I believe it is a good contribution.

================================================

At a high level, the paper considers how to sort a number of items without explicitly necessarily learning their actual meanings or values. Permutations are discrete combinatorial objects, so the paper proposes a method to perform the optimization via a continuous relaxation.

This is an important problem to sort items, arising in a variety of applications, particularly when the direct sorting can be more efficient than the two step approach of computing the values and then sorting.

I like both the theoretical parts and the experimental results. In the context of ICLR, the specific theoretical modules comprise some cute results (Theorem 4; use of past works in Lemma 2 and Proposition 5). possibly of independent interest. The connections to the (Gumbel distribution <--> Plackett Luce) results are also nicely used. This Gumbel<-->PL result is well known in the social choice community but perhaps not so much in the ML community, and it is always nice to see more connections drawn between techniques in different communities. The empirical evaluations show quite good results.

However, I had a hard time parsing the paper. The paper is written in a manner that may be accessible to readers who are familiar with this (or similar) line of research, but for someone like me who is not, I found it quite hard to understand the arguments (or lack of them) made in the paper connecting various modules. Here are some examples:

- Section 6.1 states "Each sequence contains n images, and each image corresponds to an integer label. Our goal is to learn to predict the permutation that sorts these labels". One interpretation of this statement suggests that each row of Fig 3a is a sequence, that each sequence contains n=4 images (e.g., 4 images corresponding to each digit in 2960), and the goal is to sort [2960] to [0269]. However, according to the response of authors to my earlier comment, the goal is to sort [2960,1270,9803] to [1270,2960,9803].

- I did not understand Section 2.2.

- I would appreciate a more detailed background on the concrete goal before going into the techniques of section 3 and 4.

- I am having a hard time in connecting the experiments in Section 6 with the theory described in earlier sections. And this is so even after my clarifying questions to the authors and their responses. For instance, the authors explained that the experiments in Section 6.1 have \theta as vacuous and that the function f represents the cross-entropy loss between permutation z and the true permutation matrix. Then where is this true permutation matrix captured as an argument of f in (6)? Is the optimisation/gradients in (7) over s or over the CNN parameters?

---

> ### Author Response · Authors · 2018-11-16
> **Response to reviewer questions and feedback**
>
> Thanks for reviewing our paper and the helpful feedback! We have addressed your questions and comments below.
>
> Q1. Experimental setup for Section 6.1 and Figure 3.
> A1. We can see the source of confusion now, sorry about that! We have edited the description in Section 6.1 to clarify this point and replaced what was previously Figure 3 with a more illustrative Figure 4 and a descriptive caption. The reviewer’s understanding of our last response is correct --- we have a sequence of n large-MNIST images (where each large-MNIST image is a 4 digit number) and the goal is to sort the input sequence. In Figure 4 for example, the task is to sort the input sequence of n=5 images given as [2960, 1270, 9803, 1810, 7346] to [1270, 1810, 2960, 7346, 9803].
>
> Q2. Section 2.2.
> A2. In Section 2.2, we intend to provide background on stochastic computation graphs (SCG). SCGs are a widely used tool for visualizing and contrasting different approaches to stochastic optimization, especially in the context of stochastic optimization with the backpropagation algorithm since the forward and backward passes can be visualized via the topological sorting of operators in the SCG (e.g., Figures 1, 3). Due to the lack of space, we could not include a detailed overview of stochastic computation graphs and pointed the readers to the canonical reference of Schulmann et al., 2015. The key takeaway is stated in the last paragraph of Section 2.2 --- a sort operator is non-differentiable w.r.t. its inputs and including it in SCGs necessitates the need for relaxations. For a more detailed exposition to SCGs, we have included an illustrative example in Figure 6 that grounds the terminology introduced in Section 2.2.
>
> Q3. Concrete goal in section 3 and 4.
> A3. At its core, this work seeks to include general-purpose deterministic nodes corresponding to sort operators (Section 3) and stochastic nodes corresponding to random variables defined over the symmetric group of permutations (Section 4) in computational pipelines (represented via a stochastic computation graph). Following up on the reviewer’s feedback, we have significantly expanded the motivating introductions for Section 3 and 4 to clearly state the goal beforehand and how we intend to achieve it.
>
> Q4. Connecting theory with experiments. Where is this true permutation matrix captured as an argument of f in (6)? Is the optimisation/gradients in (7) over s or over the CNN parameters?
> A4. Following up on the reviewer’s feedback, we have made the following edits in the revised version:
> - Revised Figures 4, 5 (which were Figure 3, 4 in the old version) to clearly indicate the scores “s” for each experiment.
> - Included a new Appendix E which formally states the loss functions optimized by the Sortnet approaches and explains the semantics of each terms for all three experiments.
>
> Regarding the specific follow-up questions with respect to Equation 7 and 8 (which were previously Equation 6 and 7 in the first version of the paper):
> - For the experiments in Section 6.1, the function f would include an additional argument corresponding to the true permutation matrix. We did not explicitly include the ground-truth permutation as an argument to the function f in Equation 7 to maintain the generality since such objectives also arise in unsupervised settings e.g., latent variable modeling where there is no ground-truth label. See Appendix E.1 for the precise loss function.
> - The gradients in Equation 8 are w.r.t. the scores s that parameterize a distribution q. In the experiments, the scores s are given as the output of a CNN and the optimization is over the CNN parameters. Evaluating gradients w.r.t. the CNN parameters is straightforward via the chain rule/backpropagation.
>
> Please let us know if there is any other detail that needs further clarification!

---

### Official Review · AnonReviewer1 · 2018-11-03
**An improvement to relaxed sort operators; some even-harder experiments**

**Rating:** 8
**Confidence:** 4

**Review:**

This work builds on a sum(top k) identity to derive a pathwise differentiable sampler of 'unimodal row stochastic' matrices. The Plackett-Luce family has a tractable density (an improvement over previous works) and is (as developed here) efficient to sample.

[OpenReview did not save my draft, so I now attempt to recover it from memory.]

Questions:
- How much of the improvement is attributable to the lower dimension of the parameterization? (e.g. all Sinkhorn varients have N^2 params; this has N params) Is there any reduction in gradient variance due to using fewer gumbel samples?
- More details needed on the kNN loss (uniform vs inv distance wt? which one?); and the experiment overall: what k got used in the end?
- The temperature setting is basically a bias-variance tradeoff (see Fig 5). How non-discrete are the permutation-like matrices ultimately used in the experiments? While the gradients are unbiased for the relaxed sort operator, they are still biased if our final model is a true sort. Would be nice to quantify this difference, or at least mention it.

Quality:
Good quality; approach is well-founded and more efficient than extant solutions. Fairly detailed summaries of experiments in appendices (except kNN). Neat way to reduce the parameter count from N^2 to N.

I have not thoroughly evaluated the proofs in appendix.

Clarity:
The approach is presented well, existing techniques are compared in both prose and as baselines. Appendix provides code for maximal clarity.

Originality:
First approach I've seen that reduces parameter count for permutation matrices like this. And with tractable density. Very neat and original approach.

Significance:
More scalable than existing approaches (e.g: only need N gumbel samples instead of N^2), yields better results.

I look forward to seeing this integrated into future work, as envisioned (e.g. beam search)

---

> ### Author Response · Authors · 2018-11-16
> **Response to reviewer questions and feedback**
>
> Thanks for reviewing our paper and the helpful feedback! We have addressed your questions below.
>
> Q1. How much of the improvement is attributable to the lower dimension of the parameterization? (e.g. all Sinkhorn varients have N^2 params; this has N params) Is there any reduction in gradient variance due to using fewer gumbel samples?
> A1. Precise quantification of the gains due to lower dimension of the parameterization alone is hard since the relaxation itself is fundamentally different from the Sinkhorn variants. In an attempt to get a handle on these aspects (n^2 vs. n parameters and doubly stochastic vs. unimodal matrices), we analyzed the signal-to-noise (SNR) ratio for the Stochastic Sortnet and Gumbel-Sinkhorn approaches with the same number of Gumbel samples (=5). Here, we define SNR as the ratio of the absolute value of the expected gradient estimates and the standard deviation. For the experiments in Section 6.1, the SNR ratio averaged across all the parameters is shown in Figure 8. We observe a much higher SNR for the proposed approach, in line with the overall gains we see on the underlying task.
>
> Q2. More details needed on the kNN loss (uniform vs inv distance wt? which one?); and the experiment overall: what k got used in the end?
> A2. We used a uniformly weighted kNN loss for both the Sortnet approaches, while noting that it is straightforward to extend our framework to use an inverse distance weighting. Appendix E.3 includes the formal expressions for the loss functions optimized in our framework. Furthermore, we have included new results in Table 5 which show the raw performance of Deterministic and Stochastic Sortnet for all values of k considered.
>
> Q3. The temperature setting is basically a bias-variance tradeoff (see Fig 5). How non-discrete are the permutation-like matrices ultimately used in the experiments?
> A3. That’s a great suggestion! One way to quantify the non-discreteness could be based on the element-wise mean squared difference between the inferred unimodal row stochastic matrix and its projection to a permutation matrix, for  the test set of instances. We have included these results for the sorting experiment in Table 4.
>
> Please let us know if there are any further questions!

---

> > ### Comment · AnonReviewer1 · 2018-12-11
> > **thanks for updates; looking great**
> >
> > reviewed rebuttal; still support strong accept

---

### Author Response · Authors · 2018-11-16
**Summary of the revised paper**

We thank the reviewers for their helpful comments! In light of these comments, we have revised the paper. Here is a summary of changes:
- Sections 3, 4: Motivation and background for the content in these sections have been stated more explicitly. Figure 1 has been added to supplement Section 3.
- The experimental setup in Section 6.1 and illustration in Figure 4 (which was Figure 3 in the previous version) have been revised to lend more clarity.
- Appendix E has been added to connect the experiments more concretely with the theory. This appendix includes the precise objective functions for each experiment.
- Few additional experiment analysis results (Figure 8, Table 4, Table 5) have been added.

---

### Meta-Review · Area_Chair1 · 2018-12-15
**Good paper, but writing can be improved.**

**Confidence:** 5
**Recommendation:** Accept (Poster)

**Metareview:**

This paper proposes a general-purpose continuous relaxation of the output of the sorting operator. This enables end-to-end training to enable more efficient stochastic optimization over the combinatorially large space of permutations.

In the submitted versions, two of the reviewers had difficulty in understanding the writing. After the rebuttal and the revised version, one of the reviewers is satisfied. I personally went through the paper and found that it could be tricky to read certain parts of the paper. For example, I am personally very familiar with the Placket-Luce model but the writing in Section 2.1 does not do a good job in explaining the model (particularly Eq 1 is not very easy to read, same with Eq. 3 for the key identity used in the paper).

I encourage authors to improve writing and make it a bit more intuitive to read.

Overall, this is a good paper and I recommend to accept it.